# Dietary Sphingolipids Contribute to Health via Intestinal Maintenance

**DOI:** 10.3390/ijms22137052

**Published:** 2021-06-30

**Authors:** Shinji Yamashita, Mikio Kinoshita, Teruo Miyazawa

**Affiliations:** 1Department of Life and Food Sciences, Obihiro University of Agriculture and Veterinary Medicine, Obihiro 080-8555, Japan; syamashita@obihiro.ac.jp; 2Food and Biotechnology Platform Promoting Project, New Industry Creation Hatchery Center (NICHe), Tohoku University, Sendai 980-8579, Japan; teruo.miyazawa.a7@tohoku.ac.jp

**Keywords:** ceramide, glucosylceramide, inflammation, intestine, sphingomyelin, sphingosine

## Abstract

As sphingolipids are constituents of the cell and vacuole membranes of eukaryotic cells, they are a critical component acquired from our daily diets. In the present review, we highlight the knowledge regarding how dietary sphingolipids affect our health, particularly our intestinal health. Animal- and plant-derived foods contain, respectively, sphingomyelin (SM) and glucosylceramide (GlcCer) as their representative sphingolipids, and the sphingoid base as a specific structure of sphingolipids also differs depending upon the source and class. For example, sphingosine is predominant among animal sphingolipids, and tri-hydroxy bases are present in free ceramide (Cer) from plants and fungi. Dietary sphingolipids exhibit low absorption ratios; however, they possess various functions. GlcCer facilitates improvements in intestinal impairments, lipid metabolisms, and skin disorders, and SM can exert both similar and different effects compared to those elicited by GlcCer. We discuss the digestion, absorption, metabolism, and function of sphingolipids while focused on the structure. Additionally, we also review old and new classes in the context of current advancements in analytical instruments.

## 1. Introduction

The intestine digests food materials and absorbs nutrients and water and is also deeply implicated in human health maintenance via the immune and nervous systems [1]. However, intestinal impairments, such as colon cancer and inflammatory bowel disease (IBD), are becoming increasingly problematic pathologies that affect both sexes worldwide. Despite advances in the diagnosis and treatment of these diseases, their incidence rates have steadily increased in East Asian countries, including Japan, and their incidence rates in Western countries also remain high [2,3]. Epidemiological studies indicate that intestinal impairments are strongly associated with diet [4], and dietary compounds directly interact with the colonic epithelium cells and may affect growth, differentiation, and cell death within the tissue [5]. Endogenous sphingolipids are known to play key roles in inflammation-related diseases, including intestinal impairment [6,7]. Although dietary sphingolipids are also believed to be implicated in these diseases, food-derived sphingolipids differ from endogenous sphingolipids in regard to their different structures among plants, fungi, and invertebrates and their differing routes into the body, which can include digestion and absorption [8]. In this review, we introduce sphingolipids as foods or supplements that can facilitate various functions, such as influencing intestinal health, and we focus primarily on sphingolipid structure in this context.

## 2. Diversity of Sphingolipid Classes and Base Composition in Foods

Sphingolipids are primarily located within the cell and vacuole membranes of most eukaryotes and some prokaryotes, and, based on this, we consume sphingolipids daily in our diets. Sphingolipids present in most foods primarily exist as complex sphingolipids that are composed of a sphingoid base (SB) with an amide-linked fatty acid (i.e., ceramide, Cer) and a polar head group, rather than a free Cer [8]. Throughout nature, there are diverse compositions of sphingolipid classes and sphingoid bases (Figure 1).

Sphingolipids are typically classified as neutral and acidic sphingolipids. Neutral sphingolipid classes in animal-derived foods such as meats and egg yolk primarily consist of sphingomyelin (SM; a phosphocholine as the polar head group) and also include glucosylceramide (GlcCer; a glucose) and Cer, while the acidic sphingolipid classes are gangliosides that contain more than one sialic acid within the sugar chain [9,10]. Milk and dairy products contain GlcCer and lactosylceramide (LacCer) in addition to SM, and they are rich in gangliosides [10,11]. The sphingolipid composition of offal (variety meats, e.g., brain, spinal cord, liver, and intestine) is unique. For example, the brain and spinal cord, both of which are abundant in sphingolipids, contain large amounts of galactocylceramide (GalCer) and sulfatides that are represented by GalCer 3-sulfate. The neutral sphingolipids from fish-derived foods are primarily SM and monoglycosylceramides (e.g., GalCer and GlcCer) [12,13], while foods from invertebrates such as Mollusca contain Cer phosphoethanolamine and Cer 2-aminoethylphosphonate (CAEP) instead of SM [14]. The polar head group of CAEP possesses a C-P bond that consists of a phosphorus atom that is directly bound to a carbon atom. In contrast, neutral sphingolipid classes present in foods derived from plants are primary composed of GlcCer and Cer, and rice and wheat grains contain small amounts of oligo-glycosyl Cer that possess sugar chains that are composed of glucose and mannose [15]. Plant acidic sphingolipids comprise inositol phosphoceramide (IPC) and glycosyl IPC (GIPC), both of which are markedly more abundant than is GlcCer [16]. Foods derived from fungi such as mold and mushrooms possess GlcCer, GalCer, and LacCer as neutral classes and IPC and GIPC as acidic classes [17]. Although general eukaryotes can possess higher or lower levels of GlcCer, yeasts (*Saccharomyces cerevisiae*) used for panary fermentation and sake brewing do not contain GlcCer, while yeasts contain Cer, IPC, and GIPC [18]. Additionally, sphingolipids do not exist in general prokaryotes, but Cer and the specific derivatives, such as Cer glucuronide, are present in certain prokaryotes, including acetic acid bacteria (*Acetobacter*) [19,20,21]. Fermented foods such as sake lees, a by-product of sake (rice wine) brewing, contain markedly high levels of free Cer and free SBs [22,23].

In regard to animal sphingoid bases, *trans*-4-sphingenine (sphingosine, d18:1^4*t*^) is the most prevalent, and sphinganine (d18:0) and 4-hydroxysphinganine (phytosphingosine, t18:0) also occur frequently in small amounts (Table 1). However, the small intestine, kidney, and skin, which exhibit C4-hydroxylase activity, are abundant in t18:0 [24]. Plant sphingolipids exhibit a highly diverse SB composition due to having ∆8-unsaturation and these compositions can include *trans*-8-sphingenine (d18:1^8*t*^), *cis*-8-sphingenine (d18:1^8*c*^), *trans*-4,*trans*-8-sphingadienine (d18:2^4*t*,8*t*^), *trans*-4,*cis*-8-sphingadienine (d18:2^4*t*,8*c*^), 4-hydroxy-*trans*-8-sphingenine (t18:1^8*t*^), and 4-hydroxy-*cis*-8-sphingenine (t18:1^8*c*^). Plant GlcCer possesses different base compositions depending on the plant species. The predominant bases of GlcCer are d18:2^4*t*,8*c*^ in rice and maize, d18:2^4*t*,8*t*^ in soybeans, and d18:1^8*c*^ in wheat and rye (Table 2) [25,26]. Fungal GlcCer primarily comprises the fungi-specific base, 9-methyl-*trans*-4,*trans*-8-sphingadienine (9-Me d18:2^4*t*,8*t*^) [27,28]. Cer, IPC, and GIPC primarily comprise the trihydroxy bases t18:0 and t18:1^8^ in plants and of t18:0 and 4-hydroxyicosasphinganine (t20:0) in fungi [17]. Sphingolipids derived from marine invertebrates contain unique sphingoid bases. For example, CAEP in the jumbo flying squid (*Dosidicus gigas*) primarily comprises 9-methyl-*trans*-4,*trans*-8,*trans*-10-sphingatrienine (9-Me d18:3^4*t*,8*t*,10*t*^) and hexadeca-*trans*-4-sphingenine (d16:1^4*t*^), and CAEP in the giant scallop (*Mizuhopecten yessoensis*) primarily comprises 9-Me d18:3^4*t*,8*t*,10*t*^ and 4,*trans*-8,*trans*-10-sphingatrienine (d18:3^4*t*,8*t*,10*t*^) [29,30,31,32,33].

## 3. Intake of Sphingolipids from Daily Diets

A number of researchers have investigated the sphingolipid content in diets. The daily intake of sphingolipids from the American diet has been reported as 300–400 mg according to calculations of the sphingolipid content (SM and glycosphingolipids, including GlcCer and gangliosides) in food materials [9]. The daily Japanese diet contains 130–300 mg of sphingolipids (80–220 mg of SM and CAEP; 50–80 mg of GlcCer) for young individuals and 50–80 mg of sphingolipids (10–60 mg of SM and CAEP; 30 mg of GlcCer) for elderly individuals [37]. Additionally, GlcCer intake from plants has been reported as 50 mg in the daily Japanese diet [38]. Although GIPC levels in plant materials, including cereals and vegetables, are 2- to 9-fold higher than those of GlcCer [16,23], there is still little information regarding GIPC content in the diet due to the complicated extraction and analysis of these components (GIPC is fractionated into the water layer during liquid–liquid extraction along with the organic layer). Based on this, it is likely that the actual daily intake of sphingolipids is greater than the intake levels described above.

## 4. Digestion and Absorption of Various Sphingolipids

As mentioned above, foods contain various complex sphingolipids possessing different polar head groups and SBs. Complex sphingolipids are digested to generate SBs and then absorbed via the lymph, and absorbed SBs are partially resynthesized into complex sphingolipids, while they are predominantly metabolized into fatty acids [8,9,10,32,39,40]. In the small intestinal mucosa, SM and CAEP are digested to Cer by alkaline-sphingomyelinase (alk-SMase), and glycosylceramides such as GlcCer, GalCer, and LacCer are digested into Cer by glycosylceramidase. The GIPC digestion mechanism has not yet been clarified; however, Cer-1-phosphate, which is generated from GIPC by the self-digestion of plant phospholipase D, is digested to Cer by intestinal alkaline-phosphatase [41]. Subsequently, Cer is digested to SB by ceramidase. It has been reported that undigested GlcCer and Cer are partially absorbed [42,43]. Even in the large intestine, complex sphingolipids and Cer are digested by enteric bacteria. The mechanism underlying the absorption of gangliosides is not well defined. Sialidase (neuraminidase) activity, which hydrolyzes the sialic acid residues in the ganglioside sugar chain, is much lower in the small intestinal mucosa of adult mammals than in young mammals [44]. It has been reported that GD3 (a major molecule in milk and dairy products) is not digested and is instead absorbed as an intact molecule in the human intestinal tract model, and GD3 feeding has been demonstrated to increase the levels of lipid rafts from the brush border and plasma in rats [45]. It has also been reported that GD3 is digested into Cer by endoglycosylceramidase in the large intestine based on the observed absence of intermediates (i.e., GlcCer and LacCer) [46].

Oral administration of sphingolipids results in lower digestion and absorption compared to other lipids. Additionally, the digestive ratio depends upon the sphingolipid classes and the fatty acid composition, and the absorption ratio depends upon the SB structure. Glycosphingolipids exhibit lower digestion than do phosphosphingolipids, and plant- and fungi-specific SBs exhibit lower absorption than does SB d18:1^4*t*^ from animals. Nelson previously reported that when SM with labeled d18:1^4*t*^ was administered to rats, approximately 8% and 20% of administered radioactivity levels were observed in lymph collected during a 24 h period and in feces collected for 4 days, respectively [39]. In contrast, administration of GlcCer with labeled d18:1^4*t*^ resulted in approximately 4% and 40% radioactivity found in lymph and feces, respectively [40]. The digestive enzymes easily hydrolyzed SM and GlcCer with palmitate compared to those bearing longer chains [46,47]. The absorption and metabolism characteristics of free d18:1^4*t*^ were similar to those of SM with d18:1^4*t*^. Conversely, free d18:0 exhibited a much higher absorption ratio than did free d18:1^4*t*^; however, the majority of the d18:0 absorbed into the lymph was used for triglyceride synthesis, and, based on this, the level of d18:0 incorporated into sphingolipids was almost equal to that of d18:1^4*t*^. Sugawara et al. studied an intestinal tract in vitro model and reported that plant- and fungi-specific SBs (i.e., d18:1^8*c*^, d18:1^8*t*^, d18:2^4*t*,8*c*^, d18:2^4*t*,8*t*^, 9-Me d18:2^4*t*,8*t*^, and d18:0) exhibited much lower absorption ratios as free SBs compared to those of the primary animal SB d18:1^4*t*^ due to the efflux of plant and fungal SBs by P-glycoprotein [48,49,50], and they confirmed that d18:2^4*t*,8*c*^ as the major SB in rice and maize exhibited lower absorption into rat lymph than did d18:1^4*t*^ due to efflux. Among the polyunsaturated SBs (i.e., d18:2^4*t*,8*c*^, d18:2^4*t*,8*t*^, d18:3^4*t*,8*t*,10*t*^, 9-Me d18:2^4*t*,8*t*^, and 9-Me d18:3^4*t*,8*t*,10*t*^), d18:2^4*t*,8*c*^ has been reported to be the SB that is most incorporated into generated sphingolipids in rat lymph [33].

## 5. Suppression of Intestinal Cancer by Dietary Sphingolipids

It has been reported that colon cancer alters sphingolipid metabolism [51]. Due to the decreased cellular levels of Cer that inhibit the cell cycle and induce apoptosis, a reduction in SMase and glycosylceramidase and an increase in SM synthase and GlcCer synthase are both observed in colon cancer cells. Anti-cancer agents and γ-radiation used for cancer therapy exert opposite effects on these enzymes. Dietary complex sphingolipids have been reported to exert suppressive effects on colon cancer in rodent models. To prepare these models, intraperitoneal (i.p.) injection of 1,2-dimethylhydrazine (DMH) and azoxymethane (AOM) are often used. DMH is metabolized into AOM, and this compound can act as a carcinogen, particularly in the colon. DMH induces an increase in SM levels and decrease in alk-SMase expression in the colon mucosa [51,52].

### 5.1. Sphingomyelin

Schmelz et al. characterized the chemotherapeutic effects of SM in regard to colon cancer in detail. After an interval of 1 week following DMH treatment, feeding mice a diet containing 0.1% milk-derived SM for 4 weeks alleviated the formation of aberrant crypt foci (ACF), which are the earliest visible changes involved in the formation of colon cancer, in the colons of these mice [53]. Low SM intake (0.025% and 0.05%) for 34 weeks after the interval did not affect tumor incidence in the colons of DMH-treated mice; however, it did suppress the progression of adenomas into adenocarcinomas. When comparing the milk-derived SM, synthetic SM (*N*-palmitoylsphingomyelin; C16-d18:1^4*t*^ with phosphocholine), and synthetic dihydroSM (C16-d18:0 with phosphocholine), it was observed that all SMs suppressed DMH-induced ACF formation and that the effect of dihydroSM was greater than those of the other SMs [54]. SM exhibits a therapeutic effect even if the SB structure is different, and differences in SB structures can affect the intensities. The chemopreventive and chemotherapeutic effects of SM were also investigated under conditions that included 0.05% SM feeding for 7 weeks starting from 1 week prior to DMH treatment and feeding for 45 weeks beginning from an interval of 1 week after treatment [55]. Both approaches suppressed the formation and progression of tumors and canceled DMH-inhibited apoptosis in the crypts.

For aging rats (54 weeks of age) that were treated with AOM, daily administration of SM (35 mg/kg body weight) for 6 weeks after an interval of 6 weeks suppressed ACF formation in the proximal region; however, SM feeding did not affect the AOM-mediated decrease in NK cell activity [56]. SM also suppressed ACF formation in aging rats that were not treated with AOM. To compare the effects of dietary SM in the context of *p53* deficiency, wild-type and *p53* mutant mice were treated with AOM to investigate the relationship between SM feeding and the tumor suppressor gene *p53*. The results indicated that a 0.1% SM diet for 4 weeks inhibited cell proliferation but did not induce apoptosis in the distal colon of both types of AOM-treated mice [57]. Feeding of a 0.05% SM diet for 33–38 weeks also suppressed cell proliferation in the distal colon and tumor. Loss of *p53* did not affect SM-inhibited cell proliferation.

Dietary SM exerts chemopreventive and chemotherapeutic effects in the context of colon cancer. Upon SM intake, SM and the fragments that include Cer and SB can reach the large intestine [39]. Continuous SM intake increased the activities of neutral- and alk-SMase in the small and large intestine, and SM intake reversed DMH-reduced colonic alk-SMase expression [51,52]. Therefore, the increase in the colonic levels of bioactive molecules Cer and SB is speculated to function as a protective mechanism of dietary SM.

### 5.2. Glycosphingolipids

Schmelz et al. also reported the chemotherapeutic effects of glycosphingolipids on colon cancer [46]. DMH-induced ACF formation was suppressed by 0.1% and 0.025% diets of GlcCer, LacCer, and GD3 in milk when fed for 4 weeks. Additionally, DMH-induced cell proliferation in crypts was reduced by dietary glycosphingolipids. These effects were the same as those of the milk-derived and synthetic SMs. These milk-derived sphingolipids were partially digested to generate Cer by incubation along with the colon tissue and enteric bacteria. The primary SB in milk-derived sphingolipids was d18:1^4*t*^.

We investigated the chemopreventive effects of GlcCer, and we also determined the SB structure [58]. Ten days prior to DMH treatment for 10 weeks, mice were fed 0.1% and 0.5% diets of maize-derived GlcCer that predominantly contained SB d18:2^4*t*,8*c*^ and a 0.1% diet of GlcCer from yeast (*Saccharomyces kluyveri*) that primarily contained the SB 9-Me d18:2^4*t*,8*t*^. All diets alleviated ACF formation, and the suppressive effects were essentially equal among the three groups. GlcCer bearing d18:2^4*t*,8*c*^ was found in the feces and colon mucosa of mice that were fed maize-derived GlcCer. Higher GlcCer concentrations within the diet resulted in higher GlcCer levels in the mouse feces. To clarify the suppressive mechanism of dietary GlcCer, DNA microarray and quantitative RT-PCR analyses were performed on the colon mucosa of mice that were fed 0.1% maize-derived GlcCer from 10 days prior to DMH treatment for 10 weeks [59]. GlcCer feeding increased the expression of *Soggy-1* mRNA, which suppressed the Wnt signaling pathway and decreased the expression of Ras-associated protein to induce the Ras pathway; however, no significant increases were observed in the expression of the genes, such as those of the caspase family, that induce apoptosis directly, thus suggesting that dietary GlcCer regulates cell proliferation and differentiation to prevent the development of ACF in the colon. Moreover, we investigated the effects of GlcCer feeding on colon inflammation following DMH treatment [60]. Mice were fed 0.1% maize-derived GlcCer 10 days prior to DMH treatment for 7 weeks. Antibody array analysis of inflammation-related cytokines revealed that DMH treatment increased the production of inflammatory cytokines and chemokines in the colon, while dietary maize-derived GlcCer suppressed the increased production, particularly for interferon-γ (IFN-γ)-related factors that included IFN-γ-induced protein 10 and monokine induced by IFN-γ.

When focusing on the chemopreventive mechanism of GlcCer, proliferation and apoptosis markers were examined in DMH-induced mutant crypts [61]. DMH-treated rats were fed 0.02% and 0.1% diets of rice bran-derived GlcCer that primarily contained the SB d18:2^4*t*,8*c*^ for 5 weeks from 1 week prior to DMH treatment. GlcCer feeding suppressed the colonic formation of ACF and β-catenin-accumulated crypts. Moreover, GlcCer decreased the ratio of proliferation hallmarks in ACF and β-catenin-accumulated crypts but did not affect the ratio of apoptosis hallmarks in these crypts.

Dietary soybean-derived GlcCer that primarily contains the SB d18:2^4*t*,8*t*^ has also been reported to exert chemotherapeutic effects [62]. Feeding mice 0.1% and 0.025% diets of soybean-derived GlcCer for 4 weeks suppressed DMH-induced ACF formation and the ratio of proliferation-positive cells in the crypt in the colons of these mice. Additionally, when examining mice of the multiple intestinal neoplasia (Min) strain that possess a mutation in the *adenomatous polyposis coli* (*APC*) gene, GlcCer was found to affect spontaneous tumorigenesis. Min mice were fed 0.1% and 0.025% diets of soybean-derived GlcCer for 8 weeks. Dietary GlcCer intake reduced tumor proliferation in the small intestine in a dose-dependent manner and suppressed mRNA expression of the Wnt signaling pathway-related gene *TCF4* and the angiogenesis-related gene *HIF1-α* in the intestinal mucosa.

Glycosphingolipids exert suppressive effects on colon cancer regardless of their polar head groups and SB composition. These effects may result from anti-proliferation activities via anti-inflammatory effects. Epidemiological studies have demonstrated that colon inflammation increases the incidence of colon cancer and that long-term inflammation stress induces cell dedifferentiation [2,3,4]. Glycosphingolipids exhibit lower digestion in the small intestine compared to that of phosphosphingolipids such as SM; however, they exhibit higher digestion in the large intestine that is facilitated by enteric bacteria and by intestinal digestive enzymes [39,40]. The suppressive effects on colon cancer by dietary glycosphingolipids have been observed to be similar, and dietary Cer intake was demonstrated to suppress intestinal tumor formation in Min mice, as described later [63].

### 5.3. Cer and SB

When comparing the effects of dietary Cer (*N*-palmitoylsphingosine; C16-d18:1^4*t*^), milk-derived complex sphingolipids (65% SM, 7.5% GlcCer, 20% LacCer, and 7.5% GD3 by weight), and their combination (Cer/complex sphingolipids = 40/60) on intestinal cancer using Min mice with a truncated *APC* gene product, all of these diets at 0.1% for 8 weeks were demonstrated to suppress intestinal tumor formation. Their combination exhibited the highest efficacy, and Cer exerted a better effect on the small intestine and lesser effect on the colon compared to the effects of complex sphingolipids [63]. Additionally, administration of 0.025% and 0.1% diets containing the sphingoid base analog (2S,3S,5S)-2-amino-3,5-dihydroxyoctadecane (Enigmol) for 6 weeks suppressed tumor formation in the small intestine of Min mice [64]. Dietary sphingolipids normalized abnormal β-catenin distribution and cell proliferation in the intestinal epithelial cells of Min mice. The functional components of dietary sphingolipids were confirmed to be Cer and SB. When targeting colon cancer, complex sphingolipids that can reach the colon are more effective than Cer and SB, and complex sphingolipids themselves may also facilitate intestinal protection.

### 5.4. Foods Containing Sphingolipids

To examine the effects of food materials containing sphingolipids on DMH-treated mice, mice were fed diets containing 10% dried whole milk, 10% dried maitake mushroom (*Grifola frondosa*), a 10% combination of these ingredients (each 5%), and a 20% combination of these ingredients (each 10%) from 1 week prior to DMH treatment for 10 weeks [65,66]. Milk sphingolipids comprise SM, LacCer, GlcCer, gangliosides, and Cer [9], while maitake mushroom sphingolipids comprise GlcCer, di-glycosyl Cer, Cer, and acidic sphingolipids [28]. Dried whole milk and maitake mushroom led to the suppression of ACF formation by all experimental diets; however, other parameters differed depending on the diet. Diets containing 10% milk and those containing a 20% combination markedly decreased DMH-elevated levels of TNF-α to untreated levels, while the 10% mushroom and 20% combination diets decreased the production of anti-apoptotic proteins and increased cleaved caspase-3 production. DMH treatment increased cecum pH and decreased the cecum contents of short chain fatty acids, while the 10% mushroom and 20% combination diets regulated DMH-modified cecum conditions. Diets containing milk resulted in anti-inflammation activities that were similar to those observed in a previous study on complex sphingolipids [60]; however, diets containing mushroom did not alter these activities in a detectable manner. Mushrooms and plants possessing cell walls may exhibit low availabilities of sphingolipids that form cell membranes. However, mushrooms and plants induce apoptosis of abnormal crypts [67], and, therefore, these materials are expected to exert beneficial effects in combination with animal food materials that contain high contents of sphingolipids.

To evaluate the above hypothesis that animals may not be able to utilize sphingolipids contained in foods of fungi and plant origin due to the presence of cell walls, we investigated whether dietary polished rice (RF) and its ethanol extract (RE), both of which contain the same level of GlcCer, can improve intestinal disease, and we also sought to determine if these effects depend upon the existence of GlcCer [68]. Mice were fed an RF diet (RF 150 g/kg diet) and RE diet (RE 0.5 g extracted from RF 150 g/kg diet) from 2 weeks prior to DMH treatment for 7 weeks. GlcCer contents were nearly identical in the RF and RE diets (3.0 and 2.7 mg/kg, respectively). Dietary RF and RE intake suppressed DMH-induced ACF formation, and RE in particular exhibited a significant suppressive effect. Dietary RE inhibited the DMH-induced production of almost all of the inflammation-related cytokines studied, while RF suppressed significantly fewer of these cytokines. Rice also contains Cer, GIPC, and oligo-glycosyl Cer [23,27]. It has been suggested that the lipophilic fraction containing sphingolipids in plant- and fungi-derived foods exerts protective effects against intestinal impairment; however, this fraction requires extraction, as digestion alone is not sufficient to induce its full protective action.

Subsequently, to clarify whether changes in sphingolipid composition in foods subjected to fermentation affect intestinal protection, mice were fed a sake rice extract diet (0.9 g extracted from sake rice 150 g/kg diet) or a sake lees extract diet (42.1 g extracted from sake lees 150 g/kg diet) from 2 weeks prior to DMH treatment for 7 weeks [69]. Sake lees are byproducts of brewed sake (rice wine), and the lipophilic compounds are concentrated compared to sake rice used as the raw material; however, the levels of highly polar sphingolipids, including GIPC, are markedly decreased and the levels of free Cer and free SB are markedly increased in sake lees [22,23]. Both diets suppressed DMH-induced ACF formation, the production of TNF-α and apoptosis-related proteins, and the oxidation of colon mucosa. Change in sphingolipid composition before and after fermentation are not believed to affect intestinal protection. The impacts of dietary sphingolipids on DMH-treated rodents are summarized in Table 3.

### 5.5. Apoptotic and Anti-Proliferative Effects of Sphingolipids on Colon Cancer Cells

Dietary sphingolipids are partially digested by intestinal enzymes and enteric bacteria into Cer and SB in the intestine. Exogenous Cer and SB induce apoptosis in various cancer cells, including colon cancer cells. Cer-bearing short-chain fatty acids (C2 to C8) are often used for experiments due to their cell permeability, while sphingolipids possessing long-chain fatty acids (C14 to C26) or hydroxy fatty acids are contained in foods.

The addition of d18:1^4*t*^ and d18:0 as free SB and of C2-d18:1^4*t*^ as Cer induced apoptosis and arrested the cell cycle in the G_2_/M phase in human colon cancer cells (HT29 and HCT-116), while treatment with dihydroCer (C2-d18:0) did not induce apoptosis and cell cycle arrest [71,72]. Compared to free SBs (d18:1^4*t*^, t18:0, d18:1^8*c*^, d18:1^8*t*^, d18:2^4*t*,8*c*^, d18:2^4*t*,8*t*^, d18:3^4*t*,8*t*,10*t*^, 9-Me d18:2^4*t*,8*t*^, and 9-Me d18:3^4*t*,8 *t*,10*t*^), all of the SBs used in these experiments exhibited apoptotic effects on human colon cancer cells (DLD-1, Caco-2, and WiDr), but not in normal intestinal cells [29,73,74]. SB induced a decrease in intracellular β-catenin levels in colon cancer cells. The apoptotic effect of C6-d18:1^4*t*^ was enhanced in colon cancer cells (HCT-15, HT-29, and LoVo) by treatment with P-glycoprotein inhibitors (tamoxifen, cyclosporine A, biricodar, and verapamil) [75]. Co-treatment of C6-d18:1^4*t*^ with tamoxifen increased intracellular Cer levels, decreased GlcCer levels, and induced apoptosis in a manner that was independent of *p53*. Conversely, low levels of natural Cer possessing long-chain fatty acids from the bovine brain reduced cytosolic β-catenin levels in colon cancer cells (SW480), while C2-d18:1^4*t*^ did not reduce these levels [63].

Various SBs from foods exert similar levels of apoptotic induction in colon cancer cells [29,73,74]; however, SBs with different structures exhibit the same absorption but different efflux characteristics in intestinal cells [48,49]. Therefore, SB efflux in colon cancer cells may differ from that in normal intestinal cells, and SBs may act immediately. Additionally, Cer-bearing long-chain fatty acids from foods exhibit low permeability; however, food-derived Cer may suppress the proliferating ability of colon cancer cells even at low concentrations (2.5 μM) compared to this ability of Cer-bearing short-chain fatty acids that exhibit high permeability [63]. DihydroCer-bearing short-chain fatty acids have been used in apoptosis and permeability experiments as the negative control [71,72]; however, when endogenous dihydroCer is accumulated by knockout or treatment with an inhibitor of dihydroCer desaturase, the proliferation of cancer cells is suppressed [76]. Additionally, as dietary dihydroSM treatment exerted the inhibition of the formation of ACF in DMH-treated mice compared to that of SM [54], further studies are required to confirm the effects of food-derived Cer possessing different SBs on colon cancer cells.

## 6. Suppression of Intestinal Inflammation by Dietary Sphingolipids

IBD is a form of refractory enteritis, and chronic colon inflammation increases the risk of the onset of colon cancer. Regulation of sphingolipid metabolism is an attractive therapeutic target for colitis diseases, including IBD. The colon mucosa of patients with IBD has been reported to increase the level of sphingosine kinase 1, an enzyme that phosphorylates SB to sphingosine1-phosphate (S1P) [77]. Experimental rodent models are often prepared by the administration of dextran sodium sulfate (DSS), which induces a direct injury of intestinal mucosa and inflammation via activation of T lymphocytes. Knockout or inhibitors of sphingosine kinase 1 resulted in reduced immune responses during colitis in DSS-treated mice [77,78]. Inflammatory cytokines, including TNF-α and IL1β, activated SMase activity [79] and DSS treatment increased Cer levels within the colon [80]. Deficiency in Cer synthase 6, an enzyme that generates C14- and C16-Cer from SB, protected against the development of colitis induced by an adoptive transfer method [81].

### 6.1. Sphingomyelin

Dietary SM has been reported to suppress and to stimulate colon inflammation. In a study examining the suppressive effects, mice were fed 0.1% SM (origin-unknown) from 3 days prior to 2% DSS treatment for 1 week, and SM feeding suppressed body weight loss, colon injury, and colon myeloperoxidase (MPO) activity (an indicator of neutrophil invasion) in DSS-treated mice and promoted IgA secretion into the large intestine in non-DSS-treated mice [82]. In contrast, in a study examining inflammation acceleration, mice received egg yolk-derived SM via oral gavage in combination with 2% DSS treatment for 7 d. SM doses of 4 mg or 8 mg/day (calculated as 0.1% or 0.2% of the total daily food intake) and SM administration accelerated body weight loss, colon injury, and intestinal epithelial cell apoptosis in DSS-treated mice [83]. SM administration also increased Cer content and activated caspase-3 and 9 via cathepsin D in intestinal epithelial cells. Additionally, IL-10 negative mice used as a spontaneous colitis model were gavaged with 4 mg of egg yolk-derived SM/day for 30 days, and SM administration accelerated body weight loss and colon injury.

Another study reported the effects of dietary SM on inflammation-related colon cancer induced by treatment with DSS in combination with AOM [84]. Mice with/without PPAR-γ deficiency only in epithelial and hematopoietic cells were fed a 0.1% milk-derived SM diet and received an AOM injection on the 7th day. Drinking water was changed to that containing 2% DSS on the 13th day, and DSS water was returned to normal water at the 20th day. Dietary SM suppressed colonic inflammatory lesions and subsequently shortened inflammatory recovery time, particularly in PPAR-γ expressing mice. Additionally, dietary SM increased the survival ratio and suppressed tumor formation, in both PPAR-γ-deficient and expressing mice, at 80 days after AOM injection.

Differences in the inflammation stage or SM structure may determine whether dietary SM exerts beneficial or unfavorable effects in the context of colon inflammation. Inflammation affects the intestinal barrier and sphingolipid metabolism, and Cer synthase deficiency and SMase inhibitors suppress colon inflammation in experimental colitis [81,85]. Additionally, the accumulation of endogenous Cer, especially C16-d18:1^4*t*^, accelerates inflammation via apoptosis [86,87]. In general, SM and Cer possessing long-chain fatty acids exhibit low permeability into intestinal cells; however, they can penetrate under barrier impairment. In regard to SM structure, the fatty acid composition of egg yolk-derived SM is predominantly C16, while that of milk-derived SM possesses longer chains (C23, C22, and C24), as shown in Table 1 [34]. C16-SM is more easily digested by SMase compared to SM-bearing longer chains [46]. Thus, egg yolk SM may be more easily absorbed as Cer forms into intestinal cells compared to the absorption of milk-derived SM.

### 6.2. Glycosphingolipids

To the best of our knowledge, there is only one report regarding the beneficial effects of dietary GlcCer on DSS-induced colitis [88]. Mice were fed a 0.1% maize-derived GlcCer diet at 3 days prior to DSS treatment for 14 days. Colon samples on the 5th and 15th days after the switch to 2% DSS drinking water were used for cytokine analysis and histological analysis, respectively. Dietary GlcCer exposure alleviated weight loss during DSS treatment and preserved the integrity of the colon epithelium. DSS treatment increased the colon level of MPO and the production of inflammatory cytokines and chemokines including IFN-γ, IP-10, and MIG in the colon, and dietary GlcCer suppressed their inflammatory impairment. As described above (Section 5.2), dietary maize-derived GlcCer alleviated DMH-induced colon inflammation [60]. Maize-derived GlcCer comprises 2-hydroxy long-chain fatty acids (hydroxy C20, C24, and C22), as shown in Table 2 [26], and, therefore, it may be difficult to digest and absorb as Cer forms compared to the digestion of complex sphingolipids bearing shorter chains [46,47].

Dietary ganglioside has been reported to protect against small intestinal inflammation induced by high-fat diet and lipopolysaccharide (LPS) treatment [89,90]. Rats were fed a 20% high-fat (wt%; adjusted by triglyceride) diet containing 0.02% milk-derived gangliosides (primarily GD3) for 2 weeks and then subsequently treated with i.p. LPS. Six hours after LPS treatment, dietary ganglioside exposure resulted in lower levels of IL-1β and TNF-α as inflammatory cytokines and a higher level of IL-10 as an anti-inflammatory cytokine in intestinal mucosa and plasma and suppressed the LPS-induced expression decrease in the intestinal tight junction protein occlusion and intestinal villi damage. The change in food style from a carbohydrate-based diet to a diet containing high fat has been suggested to cause an increase in IBD incidence rates in East Asia [2,3,4] due to intestinal barrier weakening and abnormal host immune responses to the enteric bacteria [6,91]. As described below (Section 7 and Section 8), dietary complex sphingolipids improve lipid metabolism, intestinal tight junctions, and enteric bacteria flora.

### 6.3. Extracts Containing Sphingolipids from Foods

The beneficial effects of a sphingolipid-rich fraction from mushrooms on DSS-induced colitis have been demonstrated [92,93]. The sphingolipid-rich fraction of mushrooms was prepared by ethanol extraction from the residue after hot-water extraction of golden oyster mushroom (*Pleurotus citrinopileatus*). Mice were fed 1% or 5% mushroom extract diets 10 days prior to DSS treatment for 25 days. Colon samples on the 18th and 26th days from the switch to 1.5% DSS drinking water were used for cytokine analysis and histological analysis, respectively. Dietary mushroom extract suppressed DSS-induced body weight loss, colon length reduction, and spleen weight. Dietary mushroom extract ameliorated colon villi damage and increased the production of inflammation-related cytokines in a dose-dependent manner. Moreover, the mushroom extract was separated into two fractions containing polar lipids (GlcCer, IPC, and GIPC) and neutral lipids (Cer and free fatty acids) through the use of acetone. Mice were fed 1% polar lipid or neutral lipid diets from 1 week prior to DSS treatment for 20 days. On the 21st day after the switch to 1.5% DSS drinking water, dietary mushroom polar lipids improved DSS-induced colon villi damage, while dietary neutral lipids caused the damage to worsen. Using differentiated Caco-2 cells as an intestinal tract in vitro model, the polar lipid fraction suppressed LPS-induced apoptosis, while treatment with the neutral lipid fraction resulted in weaker suppression. Therefore, the consumption of mushroom glycosphingolipids is speculated to contribute to colitis prevention.

### 6.4. Effects of Sphingolipids on Ex Vivo and In Vitro Inflammation

In experiments using bowel tissue obtained from infants requiring open bowel surgery for intestinal atresias, pretreatment with milk-derived ganglioside (primary species GD3) was reported to alleviate LPS-, hypoxia-, and combination-induced inflammation via suppression of inflammatory cytokine production and oxidative stress [94].

Using differentiated Caco-2 cells as a normal human intestinal model, the protective mechanism of complex sphingolipids on inflammation stress was investigated in detail. The addition of TNF-α or LPS to induce inflammatory stress decreased cell viability through the induction of apoptosis. Wheat-derived GlcCer (primarily SB d18:2^8*c*^), maize-derived GlcCer (primarily d18:2^4*t*,8*c*^), and bovine brain-derived GalCer (primarily d18:1^4*t*^) suppressed cell injury due to inflammatory stress, and there was no observed difference in these suppressive effects among all complex sphingolipids studied [95]. The production of inflammatory cytokines and chemokines induced by LPS was suppressed by maize-derived GlcCer. Additionally, exogenous maize-derived GlcCer was localized on the cell surface and not in the cytoplasm. Cer (sake lees-derived Cer and C20-d18:1^4*t*^) treatment induced apoptosis in differentiated Caco-2 cells under non-stress conditions, and Cer treatment suppressed LPS-induced apoptosis (unpublished data). In contrast, GlcCer, SBs, and S1P were not observed to induce apoptosis under non-stress conditions, and they suppressed LPS-induced apoptosis. These results suggest that GlcCer is non-cytotoxic, accumulates on cell membranes, and is metabolized during inflammation to protect intestinal cells by maintaining sphingolipid levels in cells and producing S1P. Additionally, as highly polar sphingolipids containing GIPC from mushroom suppressed LPS-induced apoptosis to a greater degree compared to the observed suppression by Cer and GlcCer [96], complex sphingolipids may also affect cell membrane functions.

Milk-derived SM suppressed LPS-induced mRNA expression of TNF-α in RAW264.7 macrophages, while SM in combination with SMase inhibitor treatment did not [97]. Cer (C16-d18:1^4*t*^ and C24-d18:1^4*t*^) and free SB (d18:1^4*t*^) also suppressed LPS-induced expression of TNF-α, while dihydroCer (C16-d18:0 and C24-d18:0) did not. In contrast, macrophages of SM synthase 2-deficient mice (deficiency of SM bearing very long-chain fatty acids) exhibited decreased sensitivity to thioglycolate and LPS [98]. The sensitivity was recovered by the addition of SM-containing long-chain fatty acids (C16 and C24), particularly C24, but not by the addition of SM with C6 and Cer with C24. Additionally, SM synthase 2-deficient mice were reported to be resistant to DSS-induced colitis [80]. These reports indicate that dietary complex sphingolipids can affect intestinal immunity.

## 7. Effects of Sphingolipids on Lipid Absorption and Energy Metabolism

Dietary sphingolipids are known to improve lipid absorption and metabolism. Oral gavage of cholesterol and other lipids inhibited their intestinal absorption into rat lymph in the presence of SM, and the effects of milk-derived SM were higher than those of egg yolk-derived SM [99]. In mice fed a 21% high-fat diet (wt%; adjusted by anhydrous milk fat), consumption of 0.25% milk-derived SM decreased serum levels of cholesterol and LPS, while consumption of egg yolk-derived SM did not [100]. In in vitro studies using differential Caco-2 cells as an intestinal tract model, the hydrolysis of SM or Cer increased the cholesterol absorption [101,102], while the addition of SB d18:1^4*t*^ reduced the cholesterol absorption and suppressed mRNA expression of the Niemann–Pick C1-Like 1 deeply implicated in cholesterol transport [103]. As shown in Table 1, the SB and fatty acid compositions of milk SM differ from those of egg yolk SM. The differences may affect SM functions via micellar solubilization and affinity with the hydrolases.

In Zucker rats with leptin functional disorder, 0.5% chicken skin-derived SM and maize-derived GlcCer diets decreased the levels of hepatic lipid and plasma non-HDL cholesterol, and, in particular, dietary GlcCer alleviated the plasma insulin reduction and adiponectin increase [35]. These results, in combination with the results of hepatic gene expression analyses, suggest that dietary sphingolipids improve insulin resistance via hepatic AMPK activation. Dietary sphingolipids were also observed to improve the insulin resistance that was induced in rats by a high-fructose diet (70 wt% instead of sucrose and starch; 0.25 mmol/kg diet of sea cucumber-derived Cer and GlcCer at 0.16 g and 0.21 g/kg diet, respectively) [104]. Interestingly, Cer exerted a stronger effect on glycogen accumulation in skeletal muscle, while GlcCer exhibited higher accumulation in the liver.

## 8. Other Beneficial Effects of Dietary Sphingolipids

There are many reports regarding the benefits of dietary sphingolipids besides those described above, and these benefits include the inhibition of tumor growth and progression in xenograft models of cancer [105,106], selective activation of enteric bacteria [100,107,108], alleviation of atopic dermatitis [109], and improvement of skin barrier functions [110,111,112,113,114]. For example, Cer and complex sphingolipids derived from various foods have been demonstrated to improve the moisture content of the skin according to human and animal studies. Using labeled sphingolipids in rodents, it was revealed that when sphingolipids are orally administered, the SB can reach the skin to be metabolized to Cer and complex sphingolipids [115,116]. According to other in vitro and in vitro studies, dietary complex sphingolipids upregulated Cer synthesis in mouse skin and free SBs derived from plants also upregulated Cer synthesis in keratinocytes compared to the upregulation observed in response to animal SB d18:1^4*t*^ [111,117]. In regard to the effects of dietary sphingolipids on enteric bacteria, Rohrhofer et al. have reviewed this in detail [118].

## 9. Conclusions

Sphingolipids are consumed during every meal. Due to their low intestinal absorption, a great deal of attention has been directed towards the direct effects of Cer and SBs as digests and bioactive molecules. Currently, it has been reported that dietary sphingolipids are beneficial to various body parts, and these effects can be due to the complex sphingolipids themselves and can be modulated by different SB structures. Sphingolipids in foods exhibit diverse structures (i.e., polar head groups, SBs, and fatty acids) depending on the source, and, therefore, it is necessary to distinguish between the functions of food-derived and endogenous sphingolipids. There is a possibility that various functions of dietary sphingolipids are exerted by intestinal homeostasis mechanisms such as nutritional absorption, intestinal barrier function, and gut immunity. Further macro- and micro-studies will further clarify the roles of dietary sphingolipids in the near future.

## Figures and Tables

**Figure 1 ijms-22-07052-f001:**
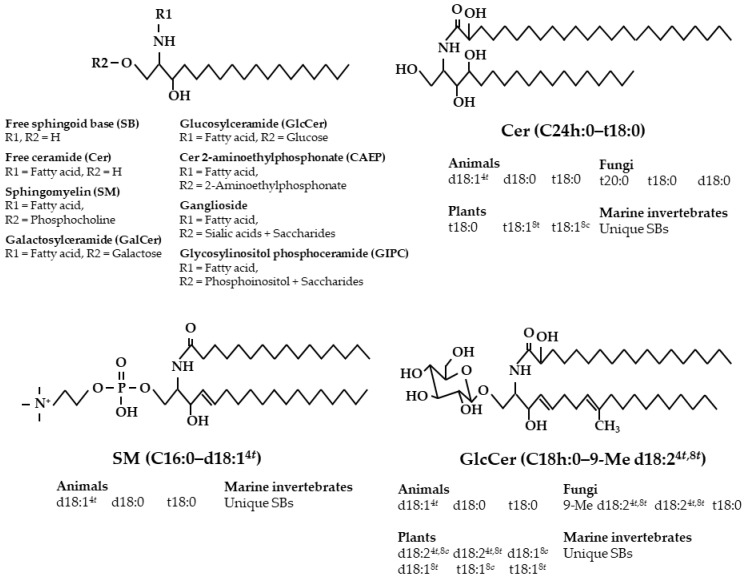
Structures of sphingolipids and diversity of sphingoid bases depend on food sources. Abbreviations: C, carbon number; *c*, *cis*; d, dihydroxy; h, α-hydroxy; Me, methyl; *t*, *trans*; t, trihydroxy.

**Table 1 ijms-22-07052-t001:** Fatty chain composition (mol%) of animal-derived sphingomyelins (SM).

		SM		
Fatty Chain	Bovine Milk [34]	Bovine Brain [34]	Egg Yolk [34]	Chicken Skin [35]
Acyl group		nonhydroxy		
C16:0	14	3	66	44
C18:0	3	42	10	20
C20:0	1	6	4	4
C22:0	22	7	6	7
C23:0	32	3	2	2
C24:0	19	6	5	6
C24:1	5	27	3	8
Others	4	6	4	9
Sphingoid base				
d16:0	9			
d17:0	15			
d17:1	8			
Me-d17:1	11			
d18:0	10	19	7	2
d18:1^4*t*^	44	82	93	98
d19:0	4			

**Table 2 ijms-22-07052-t002:** Fatty chain composition (mol%) of plant- and fungi-derived glucosylceramides (GlcCer).

	GlcCer								
Fatty Chain	Rice [26]	Wheat [26]	Rye [26]	Maize [26]	Soybean [26]	Konjac [26]	Apple [36]	Yeast [26]	Mushroom [28]
Acyl group	α-hydroxy								
C16h:0	1	39	26	6	82	19	64	1	80
C18h:0	6	7	5	16	<1	32	<1	99	12
C20h:0	50	38	36	39	<1	14	<1		
C22h:0	14	4	8	12	7	17	12		<1
C23h:0							7		2
C24h:0	21	4	7	21	8	6	15		3
C24h:1		1	8						
C26h:0	3	2	2	3	1	1			<1
Others	5	5	8	3	2	11	1		3
Sphingoid base									
d18:0	1	5	4	1	<1	<1	1		1
d18:1^4*t*^	4	1	<1	3	<1	1	<1		1
d18:1^8*c*^	<1	50	50	<1	5	3	<1		
d18:1^8*t*^	1	21	21	<1	<1	1	<1		
d18:2^4*t*,8*c*^	45	9	5	54	24	51	34		
d18:2^4*t*,8*t*^	13	4	3	17	49	11	11	20	<1
9Me-d18:2^4*t*,8*t*^								78	97
t18:0	7	1	2	2	<1	1	1	2	1
t18:1^8*c*^	26	8	12	21	12	31	36		
t18:1^8*t*^	3	1	2	2	9	1	17		

**Table 3 ijms-22-07052-t003:** Suppression and the mechanism of DMH-induced ACF formation by dietary sphingolipids.

Diet *	Animal	Diet Duration	DMH Treatment	Effects	Refs.
0.1% buttermilk- and powdered milk-derived SM	Female CF1 mice6 wks of age	For 4 wks from 1 wk	Once i.p. per wk for 6 wks	ACF formation⬇	[53]
		after final DMH i.p.	40 mg/kg bw		
0.025%, 0.05%, and 0.1% buttermilk-derived SM		For 34 wks from 1 wk	Once i.p. per wk for 6 wks	Colonic tumor incidence⬌; Adenoma progression to adenocarcinoma⬇	
		after final DMH i.p.	20 mg/kg bw		
0.1% milk-derived SM, synthetic SM (C16-d18:1^4*t*^),	Female CF1 mice	For 4wks from 1 wk	Once i.p. per wk for 6 wks	ACF formation⬇	[54]
and dihydoSM (C16-d18:0)	6 wks of age	after final DMH i.p.	40 mg/kg bw		
0.025% and 0.1% syntetic glucuronylceramide	Female CF1 mice	For 4 wks from 1 wk	Once i.p. per wk for 6 wks	ACF formation⬇	[70]
(C16-d18:1^4*t*^)	6 wks of age	after final DMH i.p.	30 mg/kg bw		
0.025% and 0.1% milk-derived GlcCer, LacCer,	Female CF1 mice	For 4 wks from 1 wk	Once i.p. per wk for 6 wks	ACF formation⬇; Cell proliferation in crypt⬇ (these data also contained	[46]
and ganglioside (GD3)	6 wks of age	after final DMH i.p.	30 mg/kg bw	0.1% milk-derived and synthetic SM diets)	
0.05% milk-derived SM	Female CF1 mice	For 7 wks from 1 wk	Once i.p. per wk for 6 wks	Colonic tumor incidence⬇; Adenoma and carcinoma⬆	[55]
	5 wks of age	prior to first DMH i.p.	30 mg/kg bw		
	Female CF1 mice	For 44 wks from 1 wk	Once i.p. per wk for 6 wks	Cell proliferation in crypt⬇; Apoptotic inhibition in crypts⬇	
	6 wks of age	after final DMH i.p.	30 mg/kg bw		
0.05% milk-derived SM based on AIN-93	Female ICR mice	For 22 wks after final	Once i.p. per wk for 6 wks	Colonic tumor formation⬇; Colonic expression and production of alk-SMase⬆;	[52]
	5 wks of age	DMH i.p.	30 mg/kg bw	Alk-SMase activity in colon mucosa⬆; Alk-SMase activity in colon content⬌	
0.1% and 0.5% maize-derived GlcCer	Male BALB/c mice	For 80 ds from 10 ds	Once i.p. per wk for 10 wks	ACF formation⬇	[58]
and 0.1% yeast-derived GlcCer	5 wks of age	before first DMH i.p.	15 mg/kg bw		
0.1% maize-derived GlcCer	Male BALB/c mice	For 80 ds from 10 ds	Once i.p. per wk for 10 wks	Colonic mRNA expression: Wnt signaling pathway suppression (Soggy-1 and others)⬆,	[59]
	5 wks of age	prior to first DMH i.p.	15 mg/kg bw	MAP-kinase pathway activation (Ras-associated protein and other)⬇, Caspase family⬌	
0.1% maize-derived GlcCer	Male BALB/c mice	For 59 ds from 10 ds	Once i.p. per wk for 7 wks	ACF formation⬇; Colonic production of inflammation-related cytokines (IP-10, MIG,	[60]
	5 wks of age	prior to first DMH i.p.	15 mg/kg bw	RANTES, I-TAC, IL-23, TNF-α, and M-CSF)⬇	
0.02% and 0.1% rice-derived GlcCer based on CE-2	Male F344 rats	For 5 wks from 1 wk	Once i.p. per 2 wks for 4 wks	ACF and BCAC formation⬇;	[61]
	42 wks of age	prior to first DMH i.p.	40 mg/kg bw	Cell proliferation in ACF and BCAC⬇; Cell apoptosis in ACF and BCAC⬌	
15% polished rice and 0.05% rice extract	Male BALB/c mice 5 wks of age	For 9 wks from 2 wks prior to first DMH i.p.	Once i.p. per wk for 7 wks 15 mg/kg bw	ACF formation⬇ only by rice extract diet (having the same GlcCer level as rice diet); Colonic production of inflammation-related cytokines (IP-10, MIG, I-TAC, TNF-α, M-CSF, and others)⬇ by rice extract diet	[68]
0.09% sake rice extract and 4.21% sake lees extract	Male BALB/c mice	For 9 wks from 2 wks	Once i.p. per wk for 7 wks	ACF formation⬇; Colonic inflammation and oxidation (TNF-α and malondialdehyde)⬇;	[69]
	5 wks of age	prior to first DMH i.p.	15 mg/kg bw	Colonic production of apoptosis-related proteins (Bcl-2, cleaved caspase-3, and others)⬇	

* Unless otherwise specified, the basal diet was AIN-76A. Abbreviations: ACF, abberant crypt foci; alk-SMase, alkaline-sphingomyelinase; BCAC, β-catenin-accumulated crypt; bw, body weight; d, day; C, carbon number; d, dihydroxy; DMH, 1,2-dimethylhydrazine; GlcCer, glucosylceramide; IFN, interferon-γ; I-TAC, IFN-inducible T cell alpha chemoattractant; i.p., intraperitoneal; IP-10, IFN-γ-induced protein 10; LacCer, lactosylceramide; M-CSF, macrophage colony-stimulating factor; MIG, monokine induced by IFN-γ; SM, sphingomyelin; RANTES, normal T cell expressed and secreted; *t*, *trans*; TNF-α, tumor necrosis factor-α; wk, week; ⬇, decrease; ⬌, no change; ⬆ increase.

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
