# Peer review of "Dietary Sphingolipids Contribute to Health via Intestinal Maintenance"

_ijms, 2021, doi:10.3390/ijms22137052_

Round 1
Reviewer 1 Report
1. line 86: "... (LacCer; a lactose) ..." --> (LacCer)
2. lines 89-90: " ...(GalCer; a galactose) ..." --> (GalCer)
3. line 88: fancy meats, like Wasyu beef?
4. line 303 & elsewhere: (IP-10) & (MIG) Remove abbreviations that are not subsequently used.
5. Table 3 is not cited in the text.
6. line 456: infection --> invasion
7. line 481: invade --> penetrate
8. line 484: C23, C20? C22:0 and C24:0?
9. lines 489-490: Suggest: To the best of our knowledge, there is only one report regarding the beneficial effects of dietary GlcCer on DSS-induced colitis.
10. lines 499-450: " ... C20, 24 and 18 ..." 18?
11. line 541: " ... [dddd] ..."
Author Response
Response to Reviewer#1
Comments:
1. line 86: "... (LacCer; a lactose) ..." --> (LacCer)
2. lines 89-90: " ...(GalCer; a galactose) ..." --> (GalCer)
4. line 303 & elsewhere: (IP-10) & (MIG) Remove abbreviations that are not subsequently used 6. line 456: infection --> invasion
6. line 481: invade --> penetrate
7. lines 489-490: Suggest: To the best of our knowledge, there is only one report regarding the beneficial effects of dietary GlcCer on DSS-induced colitis.
Response: Thanks for your helpful comments. We revised our article according to your comments.
Comment: 3. line 88: fancy meats, like Wasyu beef?
Response: Both offals and fancy meats mean edible parts other than muscle meats; e.g., brain, liver, intestine, and spleen. We revised as follows,
“The sphingolipid composition of offals (fancy meats; e.g., brain, spinal cord, liver, and intestine) is unique.”
Comment: 5. Table 3 is not cited in the text.
Response: We added a sentence as bellow.
L401-402
“The impacts of dietary sphingolipids on DMH-treated rodents are summarized in Table 3.”
Comment: 8. line 484: C23, C20? C22:0 and C24:0?
Response: We revised as follows,
“In regard to SM structure, the fatty acid composition of egg yolk-derived SM is predominantly C16, while that of milk-derived SM possesses longer chains (C23, C22, and C24), as shown in Table 1.”
Comment: 10. lines 499-450: " ... C20, 24 and 18 ..." 18?
Response: We revised as follows,
“Maize-derived GlcCer is comprised of 2-hydroxy long chain fatty acids (hydroxy C20, C24, and C22), as shown in Table 2 [26],”
Comment: 11. line 541: " ... [dddd] ..."
Response: We revised as follows,
“via suppression of inflammatory cytokine production and oxidative stress [90].”
Reviewer 2 Report
The authors in this review provide a very extensive review of the sphingolipid composition of several plants and animal-based foods, intake, and digestions of these lipids in the intestine. They highlight that in spite of the absorption rate of sphingolipids being low they play important structural and signaling roles. The authors also provided a brief overview of the role of dietary sphingolipids in the suppression of cancer, inflammation, lipid absorption, and energy metabolism. This review overall summarizes the literature available appropriately.
Author Response
Response: We express our appreciation to Reviewer#1 for the positive comments.